**Neglected Tropical Diseases**

# Surveillance and control efficacy of the Bergerac, France, 2025 chikungunya outbreak

**Sandeep Tegar**[1,2☯], **Guillaume Lacour**[3], **Antoine Mignotte**[3], **Bethan V. Purse**[1], **Christina A. Cobbold**[2], **Dominic P. Brass**[1], **Steven M. White**[1☯*]

**1** UK Centre for Ecology & Hydrology, Benson Lane, Wallingford, United Kingdom, **2** School of Mathematics and Statistics, College of Science and Engineering, University of Glasgow, Glasgow, United Kingdom, **3** Altopictus, Pérols, France

☯ These authors contributed equally to this work and share first authorship.
* smwhit@ceh.ac.uk

## Abstract:

The spread of the highly invasive mosquito, *Aedes albopictus*, across Europe, combined with climate change and human travel and trade, has led to new epidemic threats from mosquito-borne viruses, most significantly dengue and chikungunya, which are increasing in frequency and magnitude. In 2025, mainland France has seen a record number of autochthonous cases and outbreaks of chikungunya, spread across multiple locations, primarily introduced by travellers from the French Overseas Territory of La Réunion which is experiencing severe chikungunya outbreaks. Here, we describe one of the largest French outbreaks and subsequent control measures in the city of Bergerac, Dordogne, which resulted in 102 cases as of 5th November 2025. We apply a climate-driven mathematical model for *Ae. albopictus* and chikungunya virus transmission to the Bergerac 2025 outbreaks, comparing outputs to case data. The model suggests that the initial control measures in the first four weeks after the discovery of the outbreak, limited in their intervention radius and intensity, had little effect on reducing the number of cases, given the high incidence and the wide geographic extent of viral circulation. However, subsequent more widespread and intense control efforts, combined with likely increased public awareness, substantially reduced case numbers. These findings underscore the need to tailor control measures to intensity and scale of viral circulation combined with effective preventive and proactive arbovirus surveillance. Adulticides combined with public awareness campaigns can be effective for public health protection and are an important part of mitigating against the risk of *Aedes*-borne arboviruses and the ongoing outbreaks in mainland France.

## Author summary

Europe is experiencing a seismic shift in the risk of dengue and chikungunya outbreaks, vectored by the highly invasive mosquito *Aedes albopictus*. In 2025,

**Data availability statement:** All model code is available on GitHub https://doi.org/10.5281/zenodo.17534800. All case data is publicly available via Santé Publique France https://www.santepubliquefrance.fr/regions/nouvelle-aquitaine/documents/bulletin-regional/2025/chikungunya-dengue-et-zika-en-nouvelle-aquitaine.-bulletin-du-9-octobre-2025 and https://www.santepubliquefrance.fr/maladies-et-traumatismes/maladies-a-transmission-vectorielle/chikungunya/documents/bulletin-national/chikungunya-dengue-zika-et-west-nile-en-france-hexagonale.-bulletin-de-la-surveillance-renforcee-du-26-novembre-2025.

**Funding:** This study was funded by NERC IAPETUS DTP (NE/S007431/1 to SW, CC, BD & BP), EPSRC (EP/Y017838/1 and EP/Y017919/1 to CC, SW & DB) and BBSRC-Defra (BB/X018113/1 to SW & DB). The funders had no role in study design, data collection and analysis, decision to publish, or preparation of the manuscript.

**Competing interests:** The authors have declared that no competing interests exist.

mainland France experienced the largest modern European chikungunya outbreaks to date, with approximately 805 autochthonous (locally transmitted) cases of chikungunya spread across at least 81 outbreak locations. The city of Bergerac was one of the most severely affected with a total of 103 cases have been detected as of 24th November 2025. In mitigation, the local authorities attempted to control the outbreak with a vector control programme, that varied in extent and intensity. Using a state-of-the-art model, we accurately predicted the initial pattern of the outbreak and suggested that large-scale, intensive control efforts substantially reduced autochthonous case numbers by September 2025 (several weeks into the outbreak) after an epidemiological lag of approximately one week. However, initial localised control efforts had little effect on reducing case numbers, most likely due to a significant delay in detecting the outbreak. This modelling work is an important step for developing a real-time decision-support tool to help inform public health officials.

## Background

Europe is facing novel threats of human epidemic potential from mosquito-borne diseases, none more so than dengue and chikungunya, which historically were limited to tropical zones. This is due to 1) the global rise of dengue [1] and chikungunya cases [2], 2) the highly invasive mosquito, *Aedes albopictus* (Skuse, 1894) adapting to novel environments and spreading throughout Europe [3], 3) climate change affecting both mosquito fitness and their ability to transmit arboviruses [4], and 4) increased risk of global travel introducing viraemic travellers from endemic regions [5]. As a result, European outbreaks are increasing in magnitude [6,7] and frequency [8]. *Aedes*-borne disease outbreaks in Europe currently rely on importation from endemic countries often by viraemic travellers, with dengue and chikungunya being the most frequently imported [6]. These frequent imports have the potential to impact human health ranging in magnitude from sporadic cases to outbreaks [9].

In recent history (associated with the establishment vector [10] *Ae. albopictus*), mainland France has only had a small number of autochthonous chikungunya cases; from 2010 up until 2024, there were four outbreaks (2010, 2014, 2017 and 2024) with a total of 32 autochthonous cases of chikungunya reported by Santé Publique France (SPF) [11]. In contrast, in 2025, mainland France has experienced unprecedented numbers of autochthonous chikungunya cases [12]. As of 24th November 2025, there have been 805 autochthonous (locally transmitted) cases of chikungunya spread across at least 81 outbreak locations in mainland France, ranging from 1 to 141 cases per location [9]. This is most likely due to the high frequency of travel between mainland France and French Overseas territories (1,073 reported imported cases of chikungunya to mainland France as of 24th November 2025 [9]), which have experienced severe endemic outbreaks, predominantly La Réunion [9] (54,340 reported cases and 43 deaths in 2025, although the outbreak is now waning [13]). The high levels of transmission observed in 2025 in mainland France have been accompanied

by a change in epidemiology, and phylogenetic analysis suggests that, for the first time, some indigenous outbreaks have originated from another active indigenous outbreak in mainland France, rather than being imported from overseas [14]. As of 3rd October 2025, Centre National de Reference Arbovirus (CNR Arbovirus) had generated 51 chikungunya virus sequences linked to mainland autochthonous cases, all of which belong to the ECSA-2 genotype and correspond to the lineage responsible for the 2024–2025 epidemic on La Réunion [14].

The primary method for controlling arbovirus outbreaks, including dengue and chikungunya, is vector control, targeting adult mosquitoes with adulticides (pyrethroids) and juvenile life stages with larvicides (*Bacillus thuringiensis israelensis*), and eliminating larval breeding sites. This is often combined with increased public awareness in recognising symptoms, understanding transmission, personal protection and informing key audiences such as travellers and healthcare workers. However, globally, there a general paucity of data on the efficacy of control for *Aedes*-borne arboviruses. Indeed, many systematic reviews and meta-analyses have highlighted that vector control is likely to have an impact on entomological indices such as mosquito population abundance and ratio of biting mosquitoes to hosts [15–19], but there is a lack of evidence that vector control directly led to a reduction in arbovirus transmission [15–19]. As some authors suggest, the evaluation of control strategies, including whether transmission is successfully interrupted, should be of paramount importance moving forward to prevent further incursion of vector-borne pathogens and the associated public health, social and economic consequences of inaction [18].

Here, we describe the outbreak and subsequent control measures from one of the largest French 2025 chikungunya locations, namely Bergerac, Dordogne, which is the first year in which the Nouvelle-Aquitaine region has faced locally acquired cases of arboviral infections. Then, using a state-of-the-art mathematical model for chikungunya transmission by *Ae. albopictus*, we assess the effectiveness of the control measures for limiting further chikungunya cases, thus providing key evidence for public health policy and vector control decision-making.

## Methods

### Chikungunya cases in Bergerac 2025

Chikungunya is a mandatorily notifiable disease in France. Over the season where *Ae. albopictus* is active, arbovirus surveillance is implemented (for chikungunya, dengue and Zika, from 1st May through to 30th November in most French regions) to detect and manage imported cases, in order to prevent or limit autochthonous transmission [20]. Once a new autochthonous case is confirmed by laboratory analysis at CNR Arbovirus and if the individual has not recently travelled, increased surveillance and vector control is initiated to limit onward transmission [11]. This system operates reactively, only moving to control a vector population once an instance of autochthonous transmission has been confirmed. However, chikungunya can be asymptomatic, with estimates of between 17% and 40% of infections showing no symptoms [21]. This may add to a delay in the detection of infected individuals which could lead to the implementation of control measures after active viral circulation has already been established.

In Bergerac, the first case of chikungunya was confirmed on 6th August 2025. Subsequently, a total of 103 suspected (symptomatic without a test), probable (a suspected case with IGM antibodies for CHIKV) or confirmed (a suspected case with CHIKV RT-PCR or IgM and IgG antibodies for CHIKV or IgG seroconversion and two consecutive (> 10 days) samples) cases [22] have been detected as of 24th November 2025 [9], with the date of symptom onset in the index case being 23rd June 2025 [23,24] (Fig 1). Thus, there was a substantial delay in detecting the outbreak, with approximately 15 latterly identified cases occurring before the initial detection [22].

### Vector control in Bergerac 2025

Shortly after the first chikungunya case was confirmed on 6th August 2025, a vector control program commenced from 11th August 2025, initiated with the adulticide treatment (Fig 1). Thus, there was a significant gap between the onset date of

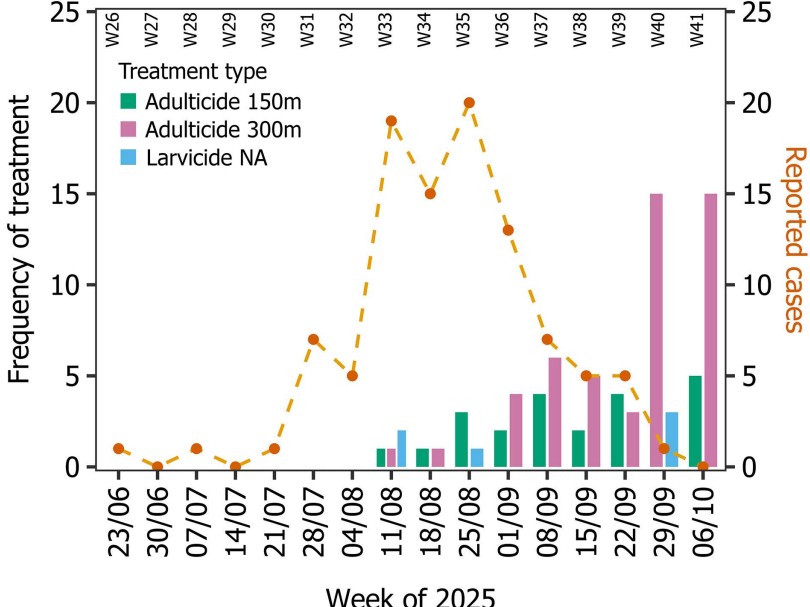

**Fig 1. Autochthonous chikungunya cases and vector control activity, Bergerac 2025.** Timeline of the autochthonous chikungunya cases (time series known as of 29th October 2025) [23,24] and vector control activity in the chikungunya outbreak region of Bergerac, 2025. Vector control is broken down by vector control treatment: adulticide over 150 m radius, adulticide over 300 m radius, and larvicide in private gardens. The frequency of adulticide over 300 m radius in weeks of 29th September and 6th October 2025 is calculated by dividing the total area from the perimeter of the treated area.

symptoms in the first known case (23rd June 2025) and the date of the first vector control intervention. This delay may be attributed to insufficient early case detection, prolonged delays in seeking medical care, late clinical diagnosis, or limitations inherent to the current surveillance and case identification system.

Between 11th August and 12th October 2025, curative vector control operations were implemented in the municipality of Bergerac in response to *Ae. albopictus* activity and the risk of chikungunya virus transmission. These interventions combined adulticidal spraying within 300 m perimeters around the likely sites of infection for confirmed human cases, together with larvicidal treatments of breeding sites in public areas and private gardens. Control decisions were based on the duration of residence of cases in the area, the suitability of the urban environment for the vector, the observed and/or trapped abundance of the vector, and the presence of other confirmed or suspected cases identified during door-to-door access by the mosquito control operator. Further, adulticidal spraying within 150 m radius was done around nearby transit areas, during the viraemic period of cases, where there was a risk of spreading the virus (Fig 1). All adulticide spraying was preceded by a door-to-door investigation in the centre of the area to obtain access to private gardens, which was necessary to prevent the vector from finding refuge from control, and by informing the residents concerned by means of a letter placed in each letterbox. Vector control activities displayed a progressive temporal intensification from mid-August (week of 11th August 2025) to early October (week of 6th October 2025), with a predominance of adulticidal spraying at increasing radii (Fig 1).

During week of 11th August 2025, initial control actions targeted two areas, mainly through larvicidal applications and limited perimeter spraying. The 300 m adulticidal spraying was repeated in the cluster in the week of 18th August 2025, all other treatments (150 m perimeters in the two subsequent weeks (n = 4)) were carried out with the aim of limiting the emergence of other clusters in the city (and outside the city: these interventions are not listed here as there was no evidence of wider arboviral circulation and not relevant to our analysis) [22].

From week 1st September 2025 onwards, the frequency and spatial extent of interventions increased markedly, reflecting a sustained response to continued epidemiological alerts. An average of 4.5±1.3 sprays of 300 m perimeter (n = 18) were realised each week between the week of 1st September to the week of 22nd September, with a first peak of activity in the week of 8th September with six 300 m perimeter adulticidal treatments (and 4 of 150 m perimeter). Comparatively, fewer treatments within a 150-metre radius were carried out during these four weeks (n = 12), as technical and human resources were prioritised for areas with confirmed circulation of chikungunya.

The following weeks (weeks of 15th and 22nd September) maintained high operational intensity, with six to seven areas treated per week, mostly by adulticidal spraying. The weeks of 29th September and 6th October 2025 marked a renewed intensification of control measures, corresponding to the largest spatial deployment of the campaign, and a change in mosquito control strategy: a 464 hectares area of Bergerac city was sprayed (excluding zones adjacent to watercourses in order to limit the impact on aquatic fauna, given the persistence of pyrethroids in water), corresponding to 15 of 300 m adulticide applications. The area was mainly treated from the road, with additional treatments in private gardens only around the most recent cases. The residents concerned were notified not by post as previously, but by two text messages sent to subscribers within the perimeter and public information via a press release posted on the town's website, as well as information stands with flyers manned by Bergerac municipal officials and the Agence régionale de santé (ARS) Nouvelle-Aquitaine at the market, the three shopping centres and the five schools in the area concerned. Three days of larvicidal actions were realised. Five 150 m spraying were realised on priority transit areas in Bergerac in the week of 6th October, following the end of the large-scale spraying of the cluster (Fig 2).

## Modelling the Bergerac 2025 chikungunya outbreak

We use a state-of-the-art arbovirus-*Ae. albopictus* model [25–27], which we summarise here (see S1 Text for further details). The mathematical framework and mosquito parameters are described in full in [25] and the chikungunya vector competence and extrinsic incubation period parameters are given in [28]. The model is a system of environmentally driven stage and phenotypically structured delay-differential equations that represent the temporal transmission of arbovirus (distinct parameterisation for either dengue or chikungunya viruses) between humans by mosquito vectors, predicting mosquito population, trait dynamics and human infections over time. This model explicitly captures the effects of environmental variation (e.g., temperature, precipitation, evaporation, photoperiod, and larval density) on mosquito and virus traits such as mosquito development rates and through-stage survival, mosquito biting rate, and the viral extrinsic incubation period (the time from taking an infected bloodmeal to a mosquito becoming infectious). We explicitly model the mosquito juvenile stages in dynamic standing water bodies that enable us to incorporate important *Ae. albopictus* behaviours, such as egg diapause and quiescence, larval competition, and larval and pupal flushing. The model is currently developed for predicting the dynamics of chikungunya or dengue outbreaks, beginning with the introduction of humans, infected with a single arbovirus strain, into a completely susceptible population. The model operates at a 2 km x 2 km resolution, which corresponds approximately with the spatial extent of the Bergerac outbreak (c.f. Fig 2).

The model has been extensively validated against independent, global datasets of historic multi-life-stage mosquito abundance and trait timeseries, as well as arbovirus outbreaks primarily vectored by *Ae. albopictus*. In each validation the model performed with high accuracy, without the need for statistically back-fitting unknown epidemiological parameters [25,27].

To model the outbreak in Bergerac, additional location specific variables are required over the outbreak timeframe as model inputs, namely environmental data and human density estimates. Daily temperature, precipitation and evaporation data were obtained from the ERA5-land climate reanalysis dataset accessed from the Copernicus Climate Data Store [https://cds.climate.copernicus.eu/] for the period 1st January 2021–19th October 2025 (44.87 degree N, 0.49 degree E, WGS84 datum). The environmental data preceding 2025 was used to allow burn-in and removal of any transient mosquito dynamics before a viraemic human is introduced to begin the transmission cycle. Human density over the outbreak region

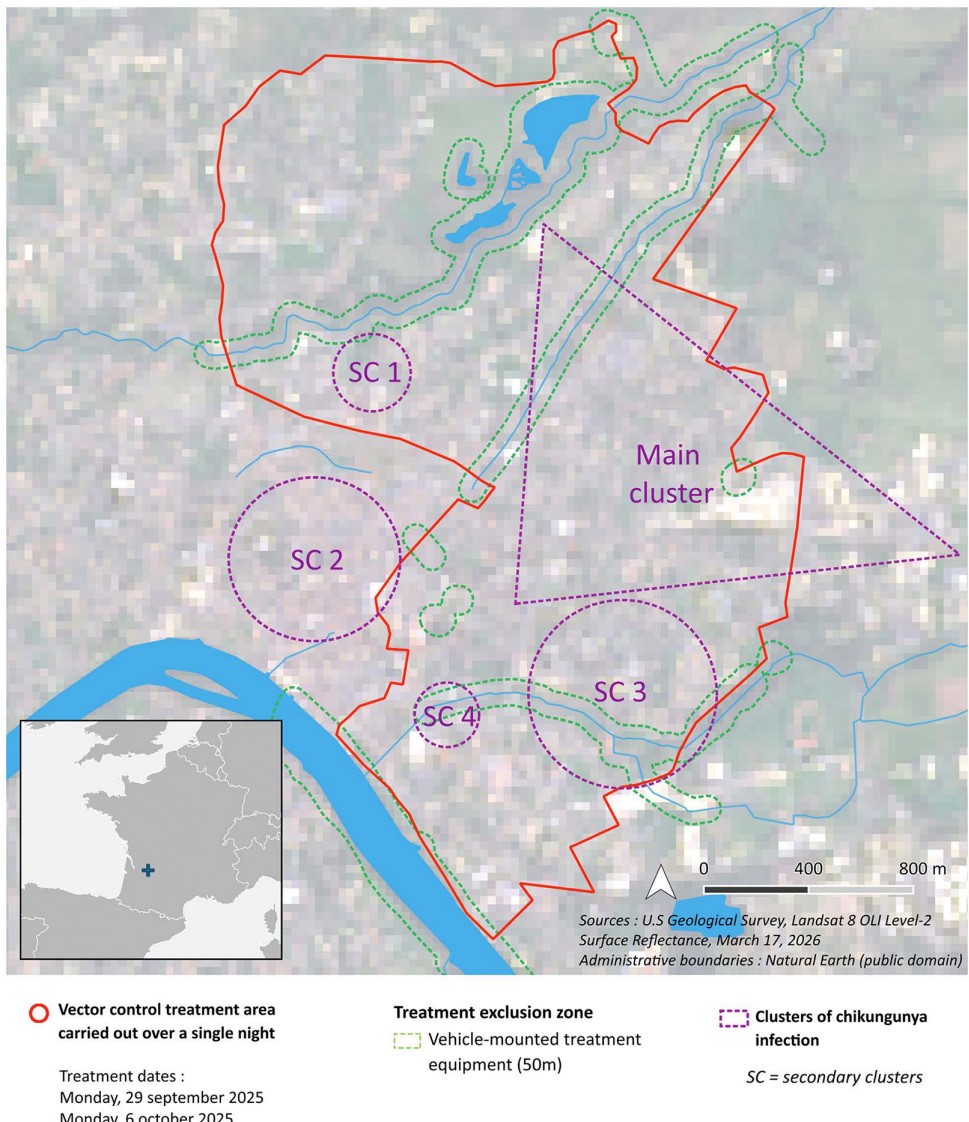

**Fig 2. Vector control treatment map.** Map of the vector control treatment area where large-scale treatments carried out over two nights in Bergerac: 29th September 2025 and 6th October 2025. Other parts of the city were concerned with adulticidal sprayings (of 150 and 300 m of radius) and are not represented on this map. Clusters of chikungunya infection are reproduced from [22]. Secondary cluster (SC) 2 was already neutralized at the date of the large-scale treatments (source: city image - Landsat http://landsat.visibleearth.nasa.gov/ and boundaries - Natural Earth https://www.naturalearth-data.com/).

in Bergerac was estimated by analysing gridded population data from the 2021 Population and Housing Census, available on the Eurostat web portal [https://ec.europa.eu/eurostat/statistics-explained/index.php?oldid=596753], resulting in a human population of 10,498 per 4 km$^2$.

Since the introduction date for the assumed viraemic traveller returning to Bergerac is unknown (the primary case), we estimate the most likely candidate for the introduction date by optimising the model to the observed case data (all other model parameters are fixed), assuming a single viraemic individual is introduced on a given day, which is varied from 15th May 2025–1st August 2025 to allow a sufficiently large period of time before the probable primary and index cases.

Optimisation is achieved numerically by minimising RMSE using case data up until 10th August 2025, before vector control was first applied to avoid bias since the mathematical model does not explicitly model control. The predicted epidemic curve (predicted weekly to align with data reporting) from the optimised primary case date and the index case date (23rd June 2025) are presented for comparison (Fig 3).

## Results

The model accurately predicts the initial phase of the epidemic curve (Fig 3), with only marginal differences between the estimate primary case date prediction (17th June 2025) and the reported index case date prediction (23rd June 2025) (RMSE = 2.03 for estimated primary date prediction; RMSE = 2.11 for reported index case date prediction). However, the models deviate substantially in August, with the estimated primary case date prediction more closely aligned with the reported case data, thus suggesting that the reported index case may not be the primary case (over the entire data reporting period, $R^2 = 0.59$ for 17th June (green) and $R^2 = 0.69$ for 23rd June (purple)). Further, since the model predictions do not substantially diverge from reported case data, the evidence would suggest that the initial control efforts had little effect on reducing cases in August.

When control efforts were intensified (week of 1st September onwards) (Fig 1) we observe substantial deviation between the modelled cases (green line) and the reported case data (orange line) (Fig 3), thus suggesting that control had a moderate effect on reducing autochthonous cases. Further, the lag between control application time and observed drop in cases is shown in the data and model comparison, noting that the model and data closely match on 8th September and deviate on 15th September, and thus aligning with the intrinsic incubation period (taken to be 1 week in the model [29] for chikungunya – black horizontal bar). Indeed, the incubation period lag between control timing and cases dropping suggests that the intensified control is likely to have an immediate effect in stopping infected mosquitoes from taking another bloodmeal and passing on their infection. The further drop in cases (21st September) suggests that the adulticide

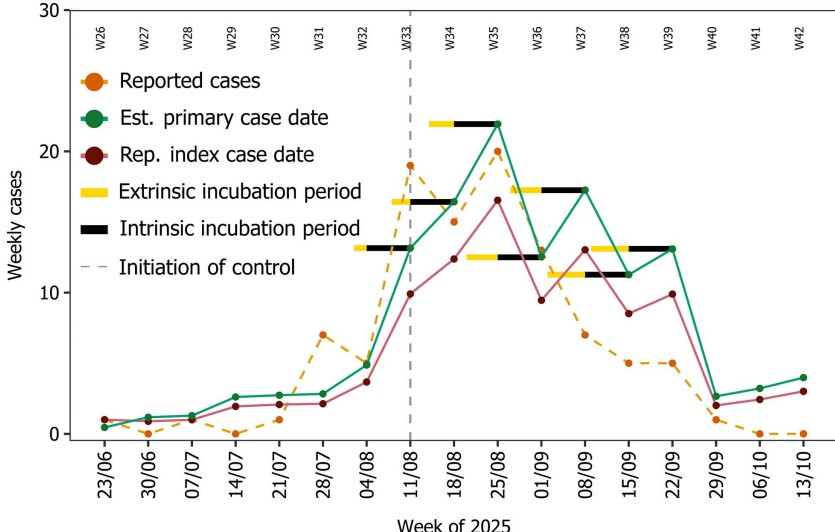

**Fig 3. Comparisons between model predictions and case data.** Outputs from the model for the Bergerac 2025 chikungunya outbreak compared to Santé Publique France case data [9,23,24]. In the plot, case data (orange, time series known as of 29th October 2025) is plotted alongside model simulations using the estimated primary case date (17th June 2025 - green) and the reported index case date (23rd June 2025 – purple). Black and yellow horizontal bars indicate the intrinsic (fixed) and the extrinsic incubation periods (variable, temperature sensitive) of chikungunya from the case reporting times in the control period, respectively. The vertical grey dashed line represents the date where vector control was initiated (Fig 1).

continued to suppress mosquito abundance and biting since the extrinsic and intrinsic incubation periods (yellow and black bar from 21st August) lie within the intensified control period.

Interventions that targeted juvenile mosquitoes, both the application of larvicides and breeding habitat management, were used less frequently during the outbreak than adulticides (breeding sites were successfully eliminated in only 75 private gardens in Bergerac). This was primarily due to the mosquito control operators' capacity being exceeded and the resulting prioritisation of adulticide measures (Fig 1). We predict that the initial larvicide interventions targeting juvenile mosquitoes would take approximately 3.5 weeks to take effect and reduce onward transmission (Fig 4 (D) – adding the larval and pupal development and the extrinsic and intrinsic incubation periods). This approximately coincides with when our model predictions and the observed case data deviate (from week of 8th September). However, given that larvicide applications were limited in their extent and frequency (Fig 1) it is most likely that larvicide treatments did not play a significant role in reducing autochthonous cases, although some moderate effects cannot be ruled out.

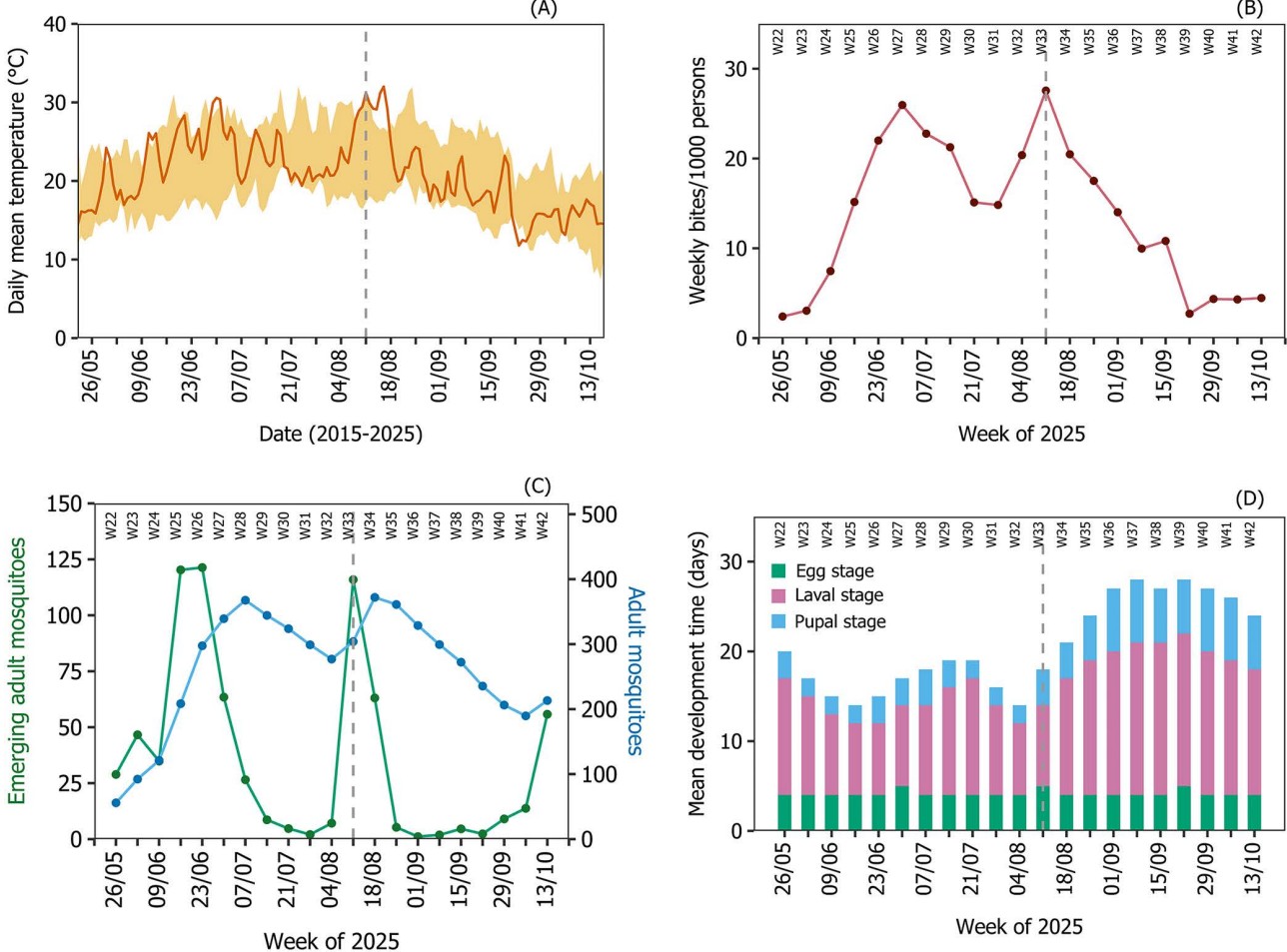

**Fig 4. Observed temperature and model outputs for Bergerac in 2025.** Panel (A) shows the observed daily mean temperature in 2025 (dark orange) and the range of historical daily mean temperatures from 2015 to 2024 (light orange ribbon) in the outbreak area. Panel (B) presents the predicted weekly number of mosquito bites per 1,000 humans in the outbreak area. Panel (C) displays the predicted weekly number of emerging adult mosquitoes (green) and the abundance of adult mosquitoes (blue). Panel (D) shows the weekly mean predicted development time for the egg, larval and pupal stages. The vertical grey dashed line represents the date where vector control was initiated (Fig 1).

Bergerac experienced unseasonally cold temperatures in the week of 22nd September (Fig 4 (A)) which likely caused a reduction in biting activity (Fig 4 (B)). This most likely contributed to the predicted drop in cases in the absence of control in the week of 29th September (Fig 3). The model also predicts, in the absence of control, an uptick in cases at the beginning of October. This is mostly likely attributable to sufficiently high temperatures from mid-September (Fig 4 (A)) causing an increase in biting (Fig 4 (B)) from the predicted pulse of newly emerged mosquitoes (Fig 4 (C)). This uptick in cases is not observed in the data (Fig 3), which is most likely attributable to the sustained intense adulticide applications (Fig 1).

The severity of the outbreak in Bergerac is mostly likely due to several interacting factors. The model shows that the timing of the primary case (most often a viraemic traveller [8]) directly affected the severity of the outbreak (S2 Fig in S1 Text) [26]. Due to the seasonal pattern of chikungunya imported from overseas into mainland France [11] which coincided with the start of the active Ae. albopictus season meant that cases were able to build up rapidly and went unreported for a substantial period (index case 23rd June; estimated primary case 17th June; first reported case 6th August) possibly due to the high rates of sub-clinical infection in people [21]. Secondly, the localised vector control efforts adopted shortly after the first case was discovered were likely to be ineffective due to inadequate coverage, especially for unknown cases. Lastly, the prolonged spike in temperatures around mid- to late August most likely caused in increase in abundance of adult mosquitoes (Fig 4 (C)) and an increase in biting pressure (Fig 4 (B)), leading to an increase in transmission.

## Discussion

In 2025, mainland France experienced the most severe outbreaks of chikungunya in modern history [2,24], with the city of Bergerac, Dordogne reporting some of the highest numbers of cases [24]. Our results suggest that large-scale, intensive control efforts moderately reduced autochthonous case numbers by September (several weeks into the outbreak) after an epidemiological lag of approximately one week. However, our results also suggest that initial limited and localised control efforts had little effect on reducing case numbers. It is probable that the late detection of the outbreak led to a rapid build-up in cases and infected mosquito populations, early in the vector activity season, such that the localised control area may not have captured all areas of circulation, allowing onward transmission. Indeed, the first spraying in Bergerac occurred 55 days after the estimated date of introduction of the primary case, a delay that likely increased the potential number of cases [30]. Elsewhere in the region, the extensive mobilisation of vector control teams around local outbreaks (16 other chikungunya outbreak locations and 1 dengue outbreak affected the Nouvelle Aquitaine region, 14 of which had been neutralised by 29th October 2025 [24]), via the dispersion of vector control efforts across numerous case transit locations and limited resources for door-to-door case search/larvicidal actions and community-awareness, probably contributed to limiting the effectiveness of the vector control measures. However, the reduction in the regional number of cases is the result of all the initiatives implemented during the outbreak, including adulticide and larvicide interventions, personal protection, and increased healthcare information and provision.

Our rapid assessment was made possible by intensive surveillance and fast data sharing [23], combined with near real-time predictions from state-of-the-art models [25–27]. However, there was a substantial period between the first case detected and the likely time the virus was introduced, which likely contributed to the severity of the outbreak [26,30]. This highlights the challenges from detecting arboviruses in the population. Currently, detection relies on raising public awareness and monitoring laboratory results, which maybe intensified during the active mosquito season [20]. However, combining existing surveillance strategies with new technologies, such as wastewater sampling, may prove beneficial for early detection of viral incursions, even when case numbers are low [31]. Once detected, current surveillance strategies seemed to work well in identify cases and targeting and implementing control strategies.

The epidemiological surveillance and vector control system in mainland France appears to be generally effective, given the limited number of cases compared to the large number of imported cases. In 2025, there were 0.76 autochthonous cases per 1 imported case of chikungunya in mainland France [24], compared to a ratio of 7.55:1 in Italy [32]. However, this may also be due to epidemiological differences, rather than solely surveillance and control differences. The

surveillance response systems may encounter difficulties in certain situations, especially when autochthonous cases go unreported, as in the situation in Bergerac, and analysing these data in conjunction model predictions may help to improve surveillance response systems. For example, rapid and accurate modelling may help to predict the potential severity of an outbreak, even with reported low case numbers [26], thus helping prioritise health and vector control resources.

The results presented here provide crucial evidence on the effectiveness of vector control methods on limiting arbovirus transmission when outbreaks are initiated – such evidence is generally lacking [15–19]. Here, we have utilised a state-of-the-art mathematical model that accurately predicts *Ae. albopictus*-borne arbovirus outbreaks [25–27]. Using only limited location-specific environmental data, the model accurately predicts the epidemic curve and suggests that the primary case date may be even earlier than the reported index case. Using such a framework it is possible to rapidly predict multiple outbreaks [26] as a near real-time decision-support tool. Further, the usefulness of the framework could be enhanced by increasing our global understanding of arbovirus importations by travellers, since varying country-level travel patterns to and from endemic arbovirus regions are likely to affect the outcomes of imported cases, and hence the outcome of locally transmitted arbovirus outbreaks.

Our interpretation on the efficacy of control stems from differences between a model (that has no control component in it) and the observed case data. Both of these constituent sources of information have likely biases that effect the uncertainty of our conclusions. For example, the model [25,26] is parameterised from laboratory data and omits some likely import environmental and socioeconomic factors, such as humidity and human movement. However, despite these shortcomings, the model has proven to be consistently accurate in predicting seasonal patterns of *Ae. albopictus* abundance and [25,27]. Case reporting biases may also have an effect, as under-reporting is common at the beginning of an outbreak which diminishes as cases and public awareness increases, as suggested by others [27,33–35]. However, correcting such biases remains an open challenge without novel and timely data.

Three main sources of uncertainty in our model predictions are the introduction date of the primary case, the human population density of the outbreak area and the spatial distribution of cases. While the model can estimate the probable date of the primary case for a given outbreak location, its predictions are highly sensitive to even small changes in this date. Similarly, the model relies on regular population grids over the outbreak area that assume a homogeneous distribution of individuals within each gird cell. In practice, however, human populations and arbovirus cases are spatially clustered, unevenly distributed, and may overlap across adjacent grid cells. This discrepancy can lead to inaccuracies in the estimated population density and host to vector ratios, although our model shows a limited degree of sensitivity to this ratio [26]. Furthermore, case data is necessarily censored to remove identifiable information, thus limiting our understanding of the outbreak, such as a lack of understanding of the spatial extent of case foci. Despite the uncertainties and sensitivities, our model consistently captures the temporal dynamics of disease outbreaks, accurately identifying their onset, peak, and decline phases [25,27]. Moreover, the model's capacity to predict dynamically changing mosquito development times, disease incubation periods, and fluctuating rates of juvenile and adult mosquito recruitment during ongoing outbreaks provides valuable insights for outbreak control and management. Such data could be predicted pre-emptively over large geographic scales using historic environmental data to highlight likely periods of mosquito actively. Previous studies on arboviral diseases in Europe have suggested that delays in case reporting can contribute to larger outbreak sizes [36], which in turn may contribute to the delays in the initiation of control measures. Furthermore, because the timing of follow-up interventions often depends on the reporting of new cases, repeated reporting delays can result in cumulative lags in the implementation of successive control efforts. In this context, our model's dynamic approach could help to optimise the timing of control programs.

There are several benefits to the approach adopted here related to good practice to public health decision-making [37], which may be of substantial value to decision-makers. For example, the model [25] has been independently developed to the case data without the need for back-fitting epidemiological and mosquito life-history parameters. The greatest source of uncertainty in the model stems from uncertainty around the date of the introduced primary case date, to which our

model is highly sensitive. However, obtaining public health information on this via passive reporting is difficult, especially when there is a prolonged delay between the first identified case and the likely introduction date. Despite this, using the reported index case the model still closely predicts the epidemiological curve and suggests that vector control was effective in reducing chikungunya transmission, albeit to a lesser efficacy, thus demonstrating robustness to our conclusions from the primary case date estimates. Previously, we have demonstrated that the model shows high accuracy across multiple independent datasets (including mosquito abundance and trait data, and epidemiological data [25,27]), suggesting that the model is robust. Lastly, no mosquito abundance dynamic data were available for the outbreak location – the presence of such data alongside the case data and the associated model comparison would have given additional confidence in our conclusions and would have offered the opportunity for refining our insights.

## Conclusions

Our results suggest that intensive, widespread use of adulticides to control *Ae. albopictus*-borne arbovirus outbreaks can be effective for protecting public health. However, the effectiveness of such approaches relies heavily on intensive and accurate surveillance. Combining near real-time data with state-of-the-art models may help decision-makers in future outbreaks.

## Supporting information

**S1 Text. Supplementary text containing full details about the model and its parametrisation, and additional results.**
(DOCX)

## Acknowledgments

The authors would like to express their sincere gratitude to all the teams and individuals who worked on epidemiological surveillance and vector control (SPF, ARS, mosquito control operators, municipal officials, etc.) in Bergerac and throughout France during the challenging context of a record epidemic year. We are grateful to Santé Publique France for providing us with the latest autochthonous case data.

## Author contributions

**Conceptualization:** Steven M. White.

**Data curation:** Guillaume Lacour, Antoine Mignotte, Steven M. White.

**Formal analysis:** Sandeep Tegar, Steven M. White.

**Funding acquisition:** Bethan V. Purse, Christina A. Cobbold, Steven M. White.

**Investigation:** Sandeep Tegar, Guillaume Lacour, Steven M. White.

**Methodology:** Sandeep Tegar, Guillaume Lacour, Antoine Mignotte, Christina A. Cobbold, Dominic P. Brass, Steven M. White.

**Supervision:** Christina A. Cobbold, Dominic P. Brass, Steven M. White.

**Validation:** Steven M. White.

**Visualization:** Sandeep Tegar, Guillaume Lacour, Antoine Mignotte, Steven M. White.

**Writing – original draft:** Sandeep Tegar, Guillaume Lacour, Antoine Mignotte, Steven M. White.

**Writing – review & editing:** Sandeep Tegar, Guillaume Lacour, Antoine Mignotte, Bethan V. Purse, Christina A. Cobbold, Dominic P. Brass, Steven M. White.

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
