## [Decision Letter · Decision Letter 0]

6 Jan 2026

Surveillance and control efficacy of the Bergerac, France, 2025 chikungunya outbreak

Dear Dr. White,

Thank you for submitting your manuscript to PLOS Neglected Tropical Diseases. After careful consideration, we feel that it has merit but does not fully meet PLOS Neglected Tropical Diseases's publication criteria as it currently stands. Therefore, we invite you to submit a revised version of the manuscript that addresses the points raised during the review process.

Please submit your revised manuscript within by March 02 2026. If you will need more time than this to complete your revisions, please reply to this message or contact the journal office at plosntds@plos.org. Please include the following items when submitting your revised manuscript:

We look forward to receiving your revised manuscript.

Kind regards,

Md. Kamrujjaman, Ph.D

Academic Editor

Amy Morrison

Section Editor

Shaden Kamhawi

co-Editor-in-Chief

Paul Brindley

co-Editor-in-Chief

**Journal Requirements:**

1) Please provide an Author Summary. This should appear in your manuscript between the Abstract (if applicable) and the Introduction, and should be 150-200 words long. The aim should be to make your findings accessible to a wide audience that includes both scientists and non-scientists. Sample summaries can be found on our website under Submission Guidelines:

Potential Copyright Issues:

- Figure S1. Please confirm whether you drew the images / clip-art within the figure panels by hand. If you did not draw the images, please provide (a) a link to the source of the images or icons and their license / terms of use; or (b) written permission from the copyright holder to publish the images or icons under our CC BY 4.0 license. Alternatively, you may replace the images with open source alternatives. See these open source resources you may use to replace images / clip-art:

**Reviewers' Comments:**

Reviewer's Responses to Questions

**Key Review Criteria Required for Acceptance?**

**Methods**

-Are the objectives of the study clearly articulated with a clear testable hypothesis stated?

-Is the study design appropriate to address the stated objectives?

-Is the population clearly described and appropriate for the hypothesis being tested?

-Is the sample size sufficient to ensure adequate power to address the hypothesis being tested?

-Were correct statistical analysis used to support conclusions?

-Are there concerns about ethical or regulatory requirements being met?

Reviewer #1: I feel they have tried their best to analyse their data after the outbreak has occurred. It is not a research project. Perhaps they are trying to do a model.

Reviewer #2: The authors used a mathematical model to evaluate the effectiveness of mosquito control activities during a chikungunya outbreak in Bergerac, France. I am not an expert in this modeling, and the applied method needs to be evaluated by an expert in this field. However, the study design the authors used is adequate to address the objectives.

Reviewer #3: The material and method require improvement to be clearer.

**Results**

-Does the analysis presented match the analysis plan?

-Are the results clearly and completely presented?

-Are the figures (Tables, Images) of sufficient quality for clarity?

Reviewer #1: This is okay. Because they are modelling after something has occurred. If they had read widely they would have noticed that there are proactive methods that have been suggested for vector surveillance especially for dengue

Reviewer #2: The results presented correspond to the analysis plan and are clearly and completely presented. The figures are ok!

Reviewer #3: The obtained results are very relevant, however the material require revision in order to interpret the results according to the real situation (biology of the species) and not only according to the model analyses.

**Conclusions**

-Are the conclusions supported by the data presented?

-Are the limitations of analysis clearly described?

-Do the authors discuss how these data can be helpful to advance our understanding of the topic under study?

-Is public health relevance addressed?

Reviewer #1: I feel more in depth analysis could have been carried out.

Reviewer #2: The conclusions of the manuscript are supported by the methods applied and the results obtained. Its relevance lies in the need to conduct studies evaluating the effectiveness of mosquito control measures during arbovirus outbreaks. It's also relevant because few studies like this have been conducted.

Reviewer #3: The conclusions follow what was presented in the results.

**Editorial and Data Presentation Modifications?**

Reviewer #1: This is not a problem

Reviewer #2: 1 - Background

- line 44: I would not say "small outbreaks". Outbreaks are outbreaks!

2 - Methods

- lines 103 to 105: to inform the case and probable case definitions for chikungunya in the French surveillance system.

- line 168: the velocity and direction of the wind could be considered as covariates in the used model?

3 - Results

- line 221: I wouldn't say a "substantial effect", instead I would say a "moderate effect".

- lines 257 to 267: I think this paragraph would be better in the Discussion item.

4 - Discussion

- line 271: I wouldn't say "substantially reduce autochthonous case numbers". I would use 'moderately' instead of 'substantially'.

Reviewer #3: (No Response)

**Summary and General Comments**

Reviewer #1: Here I feel that the authors have not read widely. It has already been shown that fogging after cases are reported are not effective. There is no correlation between cases and mosquitoes. It has been established that cases occur lag of one to 2 weeks after infected mosquitoes are found. RCT have been conducted but this has not been quoted. The authors talk about detecting the virus in wastewater. It would be more practical to detect virus in mosquitoes as simple traps can collect the adult mosquitoes. Such studies have been published since 2011.

Reviewer #2: (No Response)

Reviewer #3: Dear Authors,

Regarding the increasing risk of vector-borne diseases to the public health globally, the studies which could offer answers to the numerous questions that about it, are considered highly valuable. The large experience that countries as Italy and France gained through to fight with the autochthonous cases in the last outbreak that they faced with, represent a good source of data and lessons learnt. The manuscript titled “Surveillance and control efficacy of the Bergerac, France, 2025 chikungunya outbreak”is an interesting and very relevant material that is worth of publishing as a solid evidence of mosquito control quality assessment during the outbreak. However, there are issues that need to be addressed before the submitted material is published. First of all, the title could be significantly improved. For example: it is not control efficacy of the Bergerac but it is quality control of mosquito treatment in Bergerac or evaluation of mosquito control etc… Please be specific and precise.

Very important issues are: In the material and method section, is very difficult to follow what was done and why the procedure was changed. That should be explained in details. Why did the treatment start delayed? And why so delayed? If the outbreak was detected in June and treatment was carried out two months later, it means that you could have many, many infected mosquitoes flying around. The Ae. albopictus can survive for a long period of time (few months), so if adults avoided adulticiding they could survive the whole given period that you mentioned. And since the larvae were not in a focus, new adults were emerging again and again.

The authors were saying that larviciding did not give good effect but the operators did not perform it in the satisfying level as the authors stated. If the larviciding was not done intensively, there were always new adults coming from the water and representing new threat by spreading virus as said above.

If the procedure was not identical at the beginning of the treatments and later, we cannot compare these two periods when the treatment was done. Additionally, there is cumulative effect of treatment. If the operators did not intensively treated between June and August, and then treated intensively in August but without treating juveniles, it is clear that why will need longer period to suppress the population.

In the M&M should also be explain what data bases were used when the authors took into consideration the biology of the vector. Did you use your own data base? What about biting rates? How was that measured? Or is that estimated? Please give as many as possible details to provide good understanding and reproducibility of this study.

Please also find my specific comments:

L45 Please add year and author who described the species, when the species is mentioned for the first time in text.

L53 When you say highly competent, it is necessary to specify pathogen. Because it is not competent for all pathogens.

L57 It is not the level, but the number. Same comment for L64.

L60 Instead of “high levels of travel” it would be more suitable to say “high frequency of travelling”

L79 The authors said “many” but gave only one citation as a example. There should be give at least 5 citations if you say “many”

L96 Please replace “where” with “if” .

L115 Please explain the reasons for the gap. It is crucial for readers to understand the whole concept.

L119-124 The sentence is too long and difficult to follow. Please simplify and be more specific about what was done. From the written text it sounds like interventions represented combination of adulticiding, suitability of environment, larviciding , presence of vectors… but it is not clear what exactly was carried out.

L124 was done instead of were done

L125 …there was a risk…

L128 Explain “refuge”.

L124-131 The part about adulticiding should be specified. It is not clear whether the 150m done around confirmed case? What about 150-300?

L136 Why not?

L118 Abbreviate the species name.

L270 Cases or numbers. Delete one.

L408 Twice 2025. Please delete one.

PLOS authors have the option to publish the peer review history of their article (what does this mean?). If published, this will include your full peer review and any attached files.). If published, this will include your full peer review and any attached files.). If published, this will include your full peer review and any attached files.). If published, this will include your full peer review and any attached files.

...

Reviewer #1: No

Reviewer #2: No

Reviewer #3: No

**Figure resubmission:**
---

## [Decision Letter · Decision Letter 1]

24 Mar 2026

Dear Dr Mark White,

We are pleased to inform you that your manuscript 'Surveillance and control efficacy of the Bergerac, France, 2025 chikungunya outbreak' has been provisionally accepted for publication in PLOS Neglected Tropical Diseases.

Best regards,

Md. Kamrujjaman, Ph.D

Academic Editor

Amy Morrison

Section Editor

Shaden Kamhawi

co-Editor-in-Chief

Paul Brindley

co-Editor-in-Chief

Reviewer's Responses to Questions

**Key Review Criteria Required for Acceptance?**

**Methods**

-Are the objectives of the study clearly articulated with a clear testable hypothesis stated?

-Is the study design appropriate to address the stated objectives?

-Is the population clearly described and appropriate for the hypothesis being tested?

-Is the sample size sufficient to ensure adequate power to address the hypothesis being tested?

-Were correct statistical analysis used to support conclusions?

-Are there concerns about ethical or regulatory requirements being met?

Reviewer #1: Okay can be accepted

Reviewer #2: The authors accepted almost all the recommendations I made and adequately justified those they did not.

**Results**

-Does the analysis presented match the analysis plan?

-Are the results clearly and completely presented?

-Are the figures (Tables, Images) of sufficient quality for clarity?

Reviewer #1: can be accepted

Reviewer #2: The authors accepted almost all the recommendations I made and adequately justified those they did not.

**Conclusions**

-Are the conclusions supported by the data presented?

-Are the limitations of analysis clearly described?

-Do the authors discuss how these data can be helpful to advance our understanding of the topic under study?

-Is public health relevance addressed?

Reviewer #1: Well it only applies to Europe

Reviewer #2: The authors accepted almost all the recommendations I made and adequately justified those they did not.

**Editorial and Data Presentation Modifications?**

Reviewer #1: accept

Reviewer #2: The authors accepted almost all the recommendations I made and adequately justified those they did not.

**Summary and General Comments**

Reviewer #1: accept

Reviewer #2: The authors accepted almost all the recommendations I made and adequately justified those they did not.

PLOS authors have the option to publish the peer review history of their article (what does this mean?). If published, this will include your full peer review and any attached files.). If published, this will include your full peer review and any attached files.). If published, this will include your full peer review and any attached files.). If published, this will include your full peer review and any attached files.

...

Reviewer #1: No

Reviewer #2: No

---

## [Editor Report · Acceptance letter]

Dear Dr White,

We are delighted to inform you that your manuscript, "Surveillance and control efficacy of the Bergerac, France, 2025 chikungunya outbreak," has been formally accepted for publication in PLOS Neglected Tropical Diseases.

Best regards,

Shaden Kamhawi

co-Editor-in-Chief

Paul Brindley

co-Editor-in-Chief
